# The Middle Part of the Plucked Hair Follicle Outer Root Sheath Is Identified as an Area Rich in Lineage-Specific Stem Cell Markers

**DOI:** 10.3390/biom11020154

**Published:** 2021-01-25

**Authors:** Hanluo Li, Federica Francesca Masieri, Marie Schneider, Alexander Bartella, Sebastian Gaus, Sebastian Hahnel, Rüdiger Zimmerer, Ulrich Sack, Danijela Maksimovic-Ivanic, Sanja Mijatovic, Jan-Christoph Simon, Bernd Lethaus, Vuk Savkovic

**Affiliations:** 1Department of Cranial Maxillofacial Plastic Surgery, University Hospital Leipzig, 04103 Leipzig, Germany; hanluo.li@medizin.uni-leipzig.de (H.L.); Alexander.Bartella@medizin.uni-leipzig.de (A.B.); Sebastian.Gaus@medizin.uni-leipzig.de (S.G.); Ruediger.Zimmerer@medizin.uni-leipzig.de (R.Z.); Bernd.Lethaus@medizin.uni-leipzig.de (B.L.); 2School of (EAST) Engineering, Arts, Science & Technology, University of Suffolk, Ipswich IP 41QJ, UK; F.Masieri@uos.ac.uk; 3Clinic for Hematology, Cell Therapy and Hemostaseology, University Hospital Leipzig, 04103 Leipzig, Germany; Marie.Schneider@medizin.uni-leipzig.de; 4Polyclinic for Dental Prosthetics and Material Sciences, University Hospital Leipzig, 04103 Leipzig, Germany; Sebastian.Hahnel@medizin.uni-leipzig.de; 5Medical Faculty, Institute for Clinical Immunology, University Hospital Leipzig, 04103 Leipzig, Germany; Ulrich.Sack@medizin.uni-leipzig.de; 6Department of Immunology, Institute for Biological Research ‘Sinisa Stankovic’ (IBISS)-National Institute of Republic of Serbia, University of Belgrade, 11000 Belgrade, Serbia; nelamax@ibiss.bg.ac.rs (D.M.-I.); sanjamama@ibiss.bg.ac.rs (S.M.); 7Clinic for Dermatology, Venereology and Allergology, University Hospital Leipzig, 04103 Leipzig, Germany; Jan-Christoph.Simon@medizin.uni-leipzig.de

**Keywords:** hair follicle, outer root sheath, mesenchymal stem cells, neuroectodermal cells, non-invasive sampling, multipotency, in vitro cultivation

## Abstract

Hair follicle outer root sheath (ORS) is a putative source of stem cells with therapeutic capacity. ORS contains several multipotent stem cell populations, primarily in the distal compartment of the bulge region. However, the bulge is routinely obtained using invasive isolation methods, which require human scalp tissue ex vivo. Non-invasive sampling has been standardized by means of the plucking procedure, enabling to reproducibly obtain the mid-ORS part. The mid-ORS shows potential for giving rise to multiple stem cell populations in vitro. To demonstrate the phenotypic features of distal, middle, and proximal ORS parts, gene and protein expression profiles were studied in physically separated portions. The mid-part of the ORS showed a comparable or higher NGFR, nestin/*NES*, CD34, *CD73*, *CD44*, *CD133*, *CK5*, *PAX3*, *MITF*, and PMEL expression on both protein and gene levels, when compared to the distal ORS part. Distinct subpopulations of cells exhibiting small and round morphology were characterized with flow cytometry as simultaneously expressing CD73/CD271, CD49f/CD105, nestin, and not CK10. Potentially, these distinct subpopulations can give rise to cultured neuroectodermal and mesenchymal stem cell populations in vitro. In conclusion, the mid part of the ORS holds the potential for yielding multiple stem cells, in particular mesenchymal stem cells.

## 1. Introduction

The hair follicle outer root sheath (ORS) is recognized as a putative, non-invasively available source of adult stem cells with high developmental and regenerative potential. It is also very versatile in terms of in vitro scalable cell cultivation and multi-lineage differentiation. It harbors a heterogeneous cell pool with several putatively represented stem cell lineages, in which the distal part, named the bulge, comprises most of the stem cells. Of particular interest is a sub-population accounting for one-fifth of the cells in the bulge area, characterized as label-retaining cells (LRC) [1] and expressing a unique topological marker profile.

Among others, the bulge harbors a subpopulation of stem cells with the highest developmental potential in the adult organism named Neural Crest-like Stem Cells (NC-like SCs), which express the neural marker nestin [2,3,4,5]. NC-like SC are present mainly in the distal bulge region. These cells were initially identified in a green fluorescent protein (GFP)-knock-in mouse model [5] and later in human hair follicles [6,7,8]. Other group of multipotent stem cells have been identified in the interfollicular epidermis [9] and isthmus region [10] of the hair follicle, characterized by the expression of the transmembrane receptor Lgr6. Due to their highly naïve developmental status and the ability to generate all cell lineages of the skin [11], Lgr6+ cells are named “mother of skin cells” [10]. Next to the neural and neuroectodermal/skin stem cell lineages, the mesenchymal stem cell (MSC) lineage has also been characterized and cultured from the ORS-harvested material [12,13]. Recently, we reported a reproducible method for culturing MSCs from plucked hair follicles and upscaling them to therapy-relevant amounts [14].

Initial procedures of sampling hair follicles and establishing cell cultures were performed by extracting follicles from enzymatically processed skin rests. This helped to obtain ex vivo intact follicles, with all stem cell-rich portions. Nonetheless, this procedure requires a skin biopsy, a procedure associated with pain and possible donor-site morbidity, making it invasive and limited in clinical application. To meet the evolving requirements of personalized medicine in terms of a low-to-non-invasive approach, fewer invasive hair follicle isolation methods have also been devised and subsequently optimized. To this day, the least invasive method for obtaining cell cultures from the hair follicle has been based on explant procedure in an air-liquid interface set-up and it yielded multiple cell types [14]. The procedure has hereby bridged the gap between non-invasive ex vivo sampling and a putative in vitro culture of targeted cell types.

An obvious downside of the explant procedure based on hair plucking is a partial loss of the distal bulge part of the ORS. Due to an anatomic bond to the pyloric muscle and the sebaceous gland channel, the distal bulge portion is retained within the dermis in during follicle extraction and sampling. As a consequence, the extracted follicle contains an intact proximal and mid part, but a heavily depleted distal part [15]. Furthermore, the proximal part of the plucked follicle is a major source of dermal carry-over and poses a risk of contaminating and overpopulating the entire stem cell culture with dermal fibroblasts. Therefore, the proximal part of the follicle is routinely truncated as a part of current explant procedures [14]. In the worst case of explant procedure, the entire distal part is lost, whilst the proximal part is cut off and the mid-portion is oftenall that is left to utilize. The recently reported explant procedures for culturing keratinocytes, melanocytes, and mesenchymal stem cells resulted in successfully obtaining pure monocultures from 30–50 plucked follicles [14,16,17]. Nevertheless, those procedures utilized all the available ORS material of the plucked follicle, not exclusively an isolated central part, which is the only one that is reproducibly available.

Isolating the follicle mid-part for purposes of cell culture involves losing the bulge in the course of plucking and intentionally withdrawing the proximal follicle part by microdissection [14], based on observable morphological features. Further, the bulge is more precisely defined by the expression of specific topological markers rather than upon recognizing morphologically distinct areas. The most well-known marker for the ORS bulge is CK15; this is expressed in cells that are distributed along the ORS bulge, extending from the sebaceous gland region to the insertion site of the arrector pili muscle. Simultaneously, CD34 has been designated as a negative marker for the bulge and it is distributed between the proximal portion and the bulge area [18]. According to this map, the non-bulge part can be selected as the proximal-and-mid portion of the ORS.

Being the single reproducibly available part of plucked follicles, the none-bulge mid-ORS portion is considerably gaining importance in standardized non-invasive harvesting, not to mention its noteworthy feature for autologous personalized regenerative therapies [5,12,14]. This also provokes a query of whether the mid-ORS portion can yield a sufficient starting number of stem cells for giving rise to therapy-relevant cell numbers.

Based on these considerations, the main challenging question of this study was to determine whether the mid part of the ORS harbors the potential of giving rise to a regiment of stem cells that have formerly been isolated from intact follicles. Based on previous observation and the reported data, our working hypothesis predicted that the isolated mid-ORS part is a sufficient source of neuroectodermal, mesenchymal, and skin stem cells for scalable culturing in vitro.

The aim of the study was therefore to demonstrate this potential at the level of protein and gene expression pertaining to these major stem cell lineages, within the mid-ORS part. Moreover, here we are proposing a comprehensive analysis of differential gene expression and systematic characterization of protein marker expression patterns along the full ORS longitudinal axis, as a mean to demonstrate the potential of the three designated ORS portions, namely distal, mid and proximal, which, to our knowledge, has not been fully accomplished up to now.

## 2. Materials and Methods

Sampling of hair follicles was approved by the Ethical Committee of Medical Faculty, University of Leipzig (427/16-ek), Germany. Hair follicles in the anagen phase were plucked from donors’ temporal scalp region. 60 hair follicles were sampled randomly from the occipital regions of 7 healthy donors (*n* = 7), including 4 males with age 34.75 ± 4.55 years and 3 females with age 31 ± 2.94 years.

The follicles were washed in a medium comprised of Dulbecco’s phosphate-buffered saline (DPBS, no calcium, no magnesium) with antibiotics for 10 min (100 U/mL Penicillin, 100μg/mL Streptomycin, Sigma-Aldrich Chemie GmbH, Steinheim, DE; 50 μg/mL Gentamycin, 10 μg/mL Amphotericin B, ThermoFisher Scientific Inc., Waltham, MA, USA). Washed hair follicles were further prepared for purposes of histological characterization, flow cytometry (FCM), and qRT-PCR. For purposes of the comparative study, the hair follicles were also used to isolate mesenchymal stem cells from the plucked ORS using a migration approach, using an isolation method as detailed in Li et al. [14], and briefly described in Appendix A.

### 2.1. Histological Sections

Intact plucked hair follicles were fixed in 4% paraformaldehyde (PFA), dehydrated in ethanol gradient solutions, diaphanized in xylene (Carl Roth GmbH, Karlsruhe, DE, Germany), and embedded in paraffin. The paraffin-embedded hair follicles were longitudinally sectioned to 2 μm thickness for Hematoxylin and Eosin staining (H&E, Carl Roth GmbH, Karlsruhe, DE, Germany) and 5 μm thickness for Immunohistochemical staining.

### 2.2. Immunofluorescent Staining

Histological sections were re-hydrated and epitope retrieval was carried out at 121 °C for 16 min in a high-pressure cooker with 10 mM Sodium Citrate, 0.05% Tween 20, pH 6.0. The samples were blocked by incubation with 10% normal goat serum and incubated with primary antibodies (Table 1). Samples were washed 3 times in DPBS and subsequently incubated with secondary goat-anti-mouse or anti-rabbit IgG antibody conjugated with AlexaFluor^®^ 594 or AlexaFluor^®^ 488 (ThermoFisher Inc., Waltham, MA, USA; 1.0 mg/mL, 1:200 dilution). The nuclei were counterstained with 4′, 6-diamidino-2-phenylindole (DAPI; ThermoFisher Inc., Waltham, MA, USA; 1:400 dilution). The stained sections were observed with a Keyence BZ-9000 Fluorescence Microscope (Keyence GmbH, Neu-Isenburg, DE, Germany).

### 2.3. Immunohistochemistry Staining

For immunohistochemical analysis on the retrieved histological sections, endogenous peroxidase was blocked with 3% hydrogen peroxide, while nonspecific antibody binding o was inhibited with 10% normal goat serum for 30 min. Primary antibodies (Table 1) were diluted 1:500 and incubated for 2 h. After washing, samples were incubated with secondary antibody (Polink-1 HRP Broad Spectrum AEC/DAB Detection Kit, 1:400 dilution; GBI Labs Inc., Bothell, DC, USA) for 40 min, and incubated with Horse Radish Peroxidase-conjugated 3-amino-9-ethylcarbazole (AEC) substrate (Dako, Glostrup, Denmark) for 5–7 min or 3,3′-diaminobenzidine (DAB) substrate (Dako) for 1–2 min. The stained sections were mounted with distyrene plasticizer-xylene (DPX) Mounting Solution, and photo-documented with bright field microscopy.

### 2.4. Flow Cytometry Analysis

Intact plucked hair follicles were truncated by micro-dissecting the proximal follicle part. The processed follicles were digested using 0.04% Trypsin/0.03% EDTA (PromoCell GmbH, Heidelberg, DE, Germany) into a single cell suspension, and re-suspended in FCM buffer (DPBS containing 0.5% BSA and 0.1% NaN3) at a density of 1 × 107 cells/mL. The cells were labeled with fluorophore-conjugated antibodies (presented in Table 1), and a commercial antibody mixture was used to label negative markers of mesenchymal stem cells (PE-conjugated CD34/CD11b/CD19/CD45/HLA-DR: PE hMSC Negative Cocktail, BD Biosciences, San Jose, USA). Fluorescence intensity was analyzed by FCM (BD FACS Canto II, BD Biosciences). Gating was pre-set using the antibody fluorochrome isotype control.

### 2.5. Gene Expression

To further characterize and evaluate the phenotypic potentials amid distal, mid, and proximal portions of the ORS, plucked hair follicles were separated into these three portions by microdissection. These portions were then separately assessed using qRT-PCR to determine and differentiate gene expression of pluripotent stem cell markers (NANOG, OCT4, SOX2, NES, LGR6), mesenchymal stem cell markers (CD44, CD73, CD133, NGFR), hematopoietic cell markers (CD34, CD45, PDGFA1), keratinocyte markers (CK5, CK10, CK15) and melanocyte markers (S100, MITF, PAX3, PMEL, TYR).

Follicle segments of each portion of a single donor were pooled and analyzed as a single sample (N = 7). Hair segments were lysed in Qiazol Lysis Reagent (Qiagen, Hilden, Germany) and homogenized using QIAGEN TissueLyser II (Qiagen). Total RNA was isolated by RNeasy Plus Universal Kit (Qiagen) and quantified using NanoDrop 2000 spectrophotometer (Thermo Scientific Inc., Waltham, MA, USA). Total RNA was reverse-transcribed into cDNA using the QuantiTect Reverse Transcription Kit (Qiagen). The qRT-PCR reaction was performed using an ABI 7500 Real-Time PCR System (ThermoFisher Scientific Inc., Waltham, MA, USA) using QuantiFast SYBR^®^ Green PCR Kit (Qiagen, Hilden). Thermal cycling was set at 95 °C for 60 s, followed by 40 cycles of 95 °C for 10 s, and 60 °C for 30 s. 50 ng cDNA was used for each 20 μL reaction. Gene expression was calculated using the 2^−△Ct^ method and normalized to the housekeeping gene hypoxanthine-guanine phosphoribosyltransferase 1 (HPRT-1). Primer sequences for the assessed genes are specified in Table 2.

### 2.6. Statistical Analysis

Statistical evaluation of the quantitative results was performed by the unpaired student’s *t*-test or non-parametric Mann–Whitney test. *p* values ≤ 0.05 were regarded as statistically significant.

## 3. Results

### 3.1. Immunostaining of Longitudinal Follicle Sections

To assess the anatomic structure of the human hair follicle ORS, marker- protein expression profiles were visualized in histological sections of plucked human anagen hair follicles.

H & E staining revealed the morpho-anatomic set up of the hair follicle ORS as a compact structure with parenchymal cell content. The cuticularized IRS (inner root sheath) was observed as semitransparent, surrounding the dark brown hair shaft cortex. The dermal papilla was also visible as an intensively stained structure abundant with cells. The bulge part at the distal region of the ORS was chiefly lost in the process of plucking (Figure 1A, H&E) and its remains were not carried over with the rest of the plucked follicle.

Protein expression of Lgr6, Nestin, NGFR, CK15, PAX3, and MITF in the mid-ORS part was confirmed by the presence of a corresponding immunofluorescent signal. The proximal section of the ORS displayed a very strong signal of cells labeled with MITF (Figure 1G). The distal ORS part exhibited higher intense staining of Lgr6 and CK15 when compared to the mid-portion (Figure 1B,E,B’,E’), as well as higher PAX3 staining (Figure 1F,F’) than mid and proximal portions. Nestin was found in the inner layers of the ORS in the distal and mid-portion (Figure 1C,C’). NGFR showed the highest expression in the mid and distal part of the ORS (Figure 1D,D’).

The ORS cells in all areas showed surface and cytoplasmic expression of CD133 (Appendix A), which was clearly missing in the dermis. CD34 (Appendix A) was expressed in the outer layer of the proximal ORS region, as well as in the outer layer of the dermal sheath. There was a population of CD34-positive cells in the sub-bulge middle region of the ORS in longitudinal sections.

### 3.2. Expression of Markers Analyzed by FCM

The analysis of morphological parameters and expression of the ORS-obtained cell suspensions were initially confirmed by FACS analysis, displayed in Figure 2. Here, a heterogeneous population of cells derived from the hair follicle ORS has been observed, with varying cell sizes and antigen expression. Single-marker staining revealed cells expressing: CD44 (5.61 ± 0.70%), CD73 (32.29 ± 1.15%), CD271 (23.86 ± 1.29%), CD90 (3.06 ± 0.61%), CD105 (4.11 ± 1.52%), CD49f (Integrin alpha 6, 9.80 ± 7.40%), Cytokeratin 10 (84.07 ± 3.13%), Lgr6 (4.88 ± 0.72%), nestin (18.64 ± 2.40%), MSC Negative Markers (5.64 ± 2.92%).

A major portion of the ORS cells expressed CK10 (84.07 ± 3.13%). Considerably large portions of the cell suspension expressed CD73 (32.29 ± 1.15%) and CD271 (23.9 ± 1.39%) and nestin (18.64 ± 2.40%), whereas small portions expressed CD49f (Integrin alpha 6, 9.80 ± 7.40%), MSC markers CD44, CD90 and CD105 (5.61 ± 0.70%, 3.06 ± 0.61% and 4.11 ± 1.52%, respectively), as well as Negative MSC Markers (5.64 ± 2.92%, including CD34, CD11b, CD19 PE, CD45 PE, HLA-DR).

CD73/CD271 double-stained cells formed at least 4 identifiable populations. Two of the four populations were overlapped in Figure 1C, but remained discernable in both subpopulations 1 and 2 as shown in Appendix A. In population 1, there was a distinct cell subpopulation that was CD271+/CD73− (37.1%/52.3%, highlighted in the right lower circle); this was not observed in population 2 (CD271+/CD73−, 1.92%/44.9), as highlighted in Appendix A.

According to the forward and side scatter-based distribution (FSC/SSC) in Figure 2A,B, two major distinct cell populations could be defined in the ORS cell suspension: small, round cells and large, spindle-shaped cells (Figure 2B). Both subpopulations expressed a high level of CK10. Both populations were negative for CD105, with a marginal portion expressing CD49 at a low level (6.46% in subpopulation 1, 8.87% in subpopulation 2) as shown in Figure 2D. Based on the previous analysis of the whole cell population in Figure 2, 6.95 ± 5.88% of cells expressed CD49f and 17.47 ± 0.94% expressed D271 in the ORS cell suspension.

### 3.3. Gene Expression of Cell Type Markers Varies between Different Parts of the ORS

The distal part of the ORS showed overall higher expression levels of pluripotent stem cell marker genes compared with the mid and proximal part, including OCT4 (** *p* = 0.0074 vs. mid part), NANOG (* *p* = 0.034 vs. proximal part), SOX2 (** *p* = 0.0098 vs. mid part, ** *p* = 0.0025 vs. proximal part), LGR6 (** *p* = 0.0035 vs. mid part, *** *p* = 0.00064 vs. proximal part) and non-significantly higher levels of NES. Proximal ORS displayed a higher OCT4 expression than the mid part (** *p* = 0.002). Proximal ORS exhibited higher expression levels of CD73, a mesenchymal stem cell marker gene, when compared to the distal and mid parts (** *p* = 0.0019, * *p* = 0.024, respectively). Mid ORS also showed a significantly higher expression of NGFR, another MSC marker gene, in comparison to the proximal and distal parts (** *p* = 0.0019, * *p* = 0.024, respectively). Among the hematopoietic cell markers, the distal ORS portion displayed a reduced CD34 expression compared to the mid and proximal parts (*** *p* = 0.0006, * *p* = 0.021, respectively), yet a higher CD45 expression than the mid and proximal parts (** *p* = 0.0091, ** *p* = 0.0022, respectively). Proximal ORS exhibited a higher expression of PDGFR1 in respect to the distal and mid parts (*** *p* = 1.8 × 10−5, * *p* = 0.0013, respectively).

The mid ORS portion displayed higher levels of all keratinocyte markers CK5, CK10 and CK15 in comparison to the proximal part (*** *p* = 0.00047, * *p* = 0.022,* *p* = 0.049, respectively), with CK5 expression also superseding that of the distal part (*** *p* = 0.00012). CK10 and CK15 were expressed to the highest extent in the distal ORS part, when compared with the mid part (*** *p* = 3.08 × 10−5 vs. mid part in CK10, *** *p* = 0.00011 vs. mid part in CK15) and proximal part (*** *p* = 9.82 × 10−7 vs. proximal part in CK10, ** *p* = 0.0065 vs. proximal part in CK15).

Proximal ORS showed a higher expression of melanocyte markers, and these differences were significant for MITF (*** *p* = 0.0014 vs. distal part, ** *p* = 0.0016 vs.mid part), TYR (*** *p* = 0.0001 vs. distal part, ** *p* = 0.00196 vs.mid part) and S100 (*** *p* = 0.0012 vs. distal part). Higher expression levels of TYR and S100 were measured in the proximal ORS compared to the mid part (* *p* = 0.020 and *** *p* = 0.00033, respectively).

## 4. Discussion

This study attempted to pose and answer a question as to whether the remnant of the plucked hair follicle ORS, in particular for what concerns its middle portion, retains enough potential to give rise to different stem cell types. The present study achieved to document that the mid-ORS portion has the potential to yield neuroectodermal/NCSC-like-, mesenchymal-, hematopoietic-, keratinocyte- and melanocyte stem cells.

It was reported that the parts of the plucked hair follicle remaining available after plucking and processing procedures were able to yield graft- and therapy-relevant numbers of melanocytes, keratinocytes, and MSCs [14,16,19,20]. In reality, major ORS bulge portions are lost during plucking; the proximal part is discarded to avoid contamination and only the mid part remains intact and can be reproducibly harvested. Given these intrinsic limitations, we pointed our attention to the mid part of the ORS as the target area of our standard operative procedure for stem cell isolation based on explant protocol and air-liquid interface culture.

To this end, by the means of immunofluorescence, we have shown that the expression of neuroectodermal markers nestin, Lgr6, NGFR, mesenchymal marker CD44 [21], keratinocyte/bulge marker CK15, melanocyte markers PAX3 and MITF (Figure 1), CD133 (Appendix A) can remarkably all be allocated, among others, to the mid-ORS portion of the plucked follicle.

Overall, the immunostaining of longitudinal sections of the ORS indicated specific populations of pluripotent to multipotent stem cells, mesenchymal stem/stromal cells, hemopoietic, keratinocyte, and melanocyte progenitors in the hair follicle ORS, as shown through Lgr6, nestin, NGFR, CK15, PAX3 and MITF expression (Figure 1). Accordingly, some identified portions of the whole ORS expressing CK10, CD73, and CD271 as analyzed by FACS (Figure 2), supported the finding that the ORS may contain a presumptive subpopulation of neuroectodermal, ectodermal and mesenchymal stem cells (Appendix A). On a deeper analysis, however, it can be argued that it is possible to derive a conglomerate of small cell subpopulations, whereby two better represented specific subpopulations can be further separated (Appendix A, scatterplot panel).

In an attempt to clarify the two distinctive cell morphologies observed in the single-cell suspension, a subpopulation of ORS with a generally higher potential for yielding stem cells was pinpointed. The cells were sorted based on their Forward Scatter/Side Scatter in FACS plots and re-analyzed (Appendix A). As pointed out, both cell populations showed predominant expression of CK10. However, there was a small subset in cell population 1 that was negative for CK10 (Appendix A). In contrast with the double-stained cells identified in Figure 2C, we speculate that these cells could possibly be neuroectodermal stem cells within the ORS single-cell suspension characterized by the expression of CD271+/nestin+/CD49f+/CK10−. In turn, this small but potentially interesting cell sub-population clearly differs from keratinocytes, as they lack CK10 expression. This subset can be traced back to the subpopulation of cells that have morphologically been characterized as small, round cells, as opposed to the larger, spindle, shaped cells.

Amid these mentioned stem cell markers, nestin and Lgr6 gained utmost focus as markers of the lineages with the highest developmental potential in the ORS [22]. The longitudinal section marker distribution analysis highlighted a nestin-positive cell population in the middle and lower part of the ORS (Figure 1D), which is also in agreement with previously reported research by Li et al. [23]. As the marker of bulge stem cells [24], Lgr6 was identified by its intensive immunofluorescent signal in the upper and mid portions of the plucked ORS (Figure 1C), whereas its gene expression was concentrated in the upper part of the ORS, as expected from historical data [10].

Nestin+ cells were identified in FCM analysis with an abundance of 18.64 ± 2.40%. Only 4.88 ± 0.72% were Lgr6+ positive, as shown in Figure 2. Double labeling revealed that almost all the Lgr6+ cells expressed nestin as well, which accounted for 4.41 ± 0.66% of nestin+/Lgr6+ cells in the whole trypsinized cell population, derived from the mid-ORS. This low abundance suggests that bulge and proximal portion depletion severely reduces the potential to culture neuroectodermal lineages from explanted hair follicle, but the potential to give rise to epidermal stem cell and skin lineages is retained, and it qualifies asa viable option to follow in research as well as translationally.

FCM analysis of the plucked follicle ORS revealed that the cells with the highest developmental potency were seriously depleted by the loss of the distal bulge part.

To further assess the presence of MSCs in the ORS cell pool of plucked follicles, CD49f, CD105, CD73, and CD271 were used as markers [25] in line with previously reported studies [8,26]. FCM double staining was determined based on previously reported methods, specifically CD105/CD49f [27] and CD73/CD271 [28,29,30,31,32,33,34,35,36].

FCM analysis confirmed that 9.80 ± 7.40% of CD49f+ cells and 4.11 ± 1.52% of CD105+ cells were found in the mid-ORS, whereas the part of ORS cells simultaneously expressing CD49f+/CD105+ was about 2.85 ± 1.57%, which is comparable with the 4.52% previously reported by Yang et al. [25].

The number of both CD73- and CD271-expressing cells in the mid-ORS pool was considerably high: 32.29 ± 1.15% and 23.86 ± 1.29%, respectively. Double staining of CD73/CD271 showed 8.39 ± 0.50% of co-expressing cells. Interestingly, in both subpopulations, there were 4 groups of cells identified from the double-staining profile of CD73/CD271 (Figure 2C and Appendix A). MSC populations that co-expressed CD271 with CD73 were previously identified in a low abundance of less than 10% in BMMSCs and in Adipose-derived MSCs (ADMSCs) [37,38,39], in line with the 8% portion of CD271+/CD73+ cells found in the mid-ORS. Probably, this subpopulation may be sufficient to generate a primary culture of pure Mesenchymal Stem Cells derived from the ORS (MSCORS), which eventually would express MSC markers in the highest proportion (Figure 2 and Appendix A) and can robustly be differentiated towards the classic mesenchymal tri-lineage differentiation, yielding osteoblast, adipocyte and chondroblast-like cells (Appendix A) [14].

Differential gene expression across the ORS was also assessed by micro-dissecting the three portions (proximal, mid, and distal) and subsequently analyzing gene expression in each portion. Differential analysis displayed that the cells expressing neuroectodermal-, MSC-, keratinocyte- and melanocyte stem cell markers were distributed in the mid-ORS part as well. Both protein and gene expression patterns of CK15 and CD34 in the relevant ORS portion correlated with previously reported [18,40], high levels of the bulge marker CK15 expressed in the distal ORS, whereas proximal ORS and epithelial cells of the external root sheath expressed the negative bulge marker CD34 (Figure 1 and Appendix A) [41]. On the other hand, CD271/NGFR was expressed in the mid part in both histology (Figure 1) and qRT-PCR (Figure 3), identified in 23.86 ± 1.29% of ORS cells (Figure 2E). Gene expression pattern of melanocyte markers MITF, TYR, and S100 can accordingly be allocated primarily to the proximal ORS part, with an evenly distributed expression of PAX3 and PMEL across the three portions.

Notably, the mid-ORS cell population contained two putative portions of specific stem cells (CK10+ and CD73+ cells), whereas the rest of the subpopulations were represented in lower abundance. Nevertheless, the portion of double-stained Lgr6+/nestin+ cells, known NCSC-like stem cell marker, was 4.41 ± 0.66%. Moreover, CD73/CD271 cells, matching the MSCs, reached a proportion of 8.39 ± 0.50%, which remains in line with the previously reported portion of label-retaining cells in the bulge [1].

The aforementioned findings show that the remaining potential of the middle ORS portion to give rise to stem cells after plucking is consistent with the presence of NCSC-like/neuroectodermal, mesenchymal, and melanocyte stem cells. In particular, the subpopulation designated as MSCs, situated in the mid part of the ORS, is apparently sufficient to warrant cell migration, proliferation and for giving rise to a stable MSCORS culture that can be up-scaled to millions of cells [14].

This potential appears to rely on the isolation method using air-liquid interface migration of the ORS. If directly plated onto the cell culture flask, cells from ORS single-cell suspension failed to adhere as shown in Appendix A. The marked difference in cell morphology was observed in that MSCORS were able to adhere to a polystyrene uncoated culture surface and exhibited bipolar/multipolar elongated morphology. Further, we observed high expression of MSC markers CD73, CD90, CD105 combined with CD44 and no expressions of MSC negative markers. These data are consistent with confirming the mesenchymal/stromal nature of these cells [14]. To further investigate the nature and capacity for differentiation of ORS-derived mesenchymal stem/stromal cells, these were differentiated successfully towards the typical three lineages: osteogenic, adipogenic, and chondrogenic, as shown in Appendix A, therefore confirming their multipotent capacity [25,42]; this was done as previously described by Li et al. 2020 [14]. Using the same plucked hair follicle ORS, the cell populations obtained using primary digestion and MSCORS isolation method yielded distinctively different outcomes.

## 5. Conclusions

Collectively, the results of this study indicate that the potential of the ORS to yield neuroectodermal, mesenchymal, keratinocyte, and melanocyte stem cells can be allocated to its readily available mid-portion and that the mid-ORS part can also be used as a sufficient and reproducible source for culturing those cells, in particular, the skin- and mesenchymal stem cell lineage.

## Figures and Tables

**Figure 1 biomolecules-11-00154-f001:**
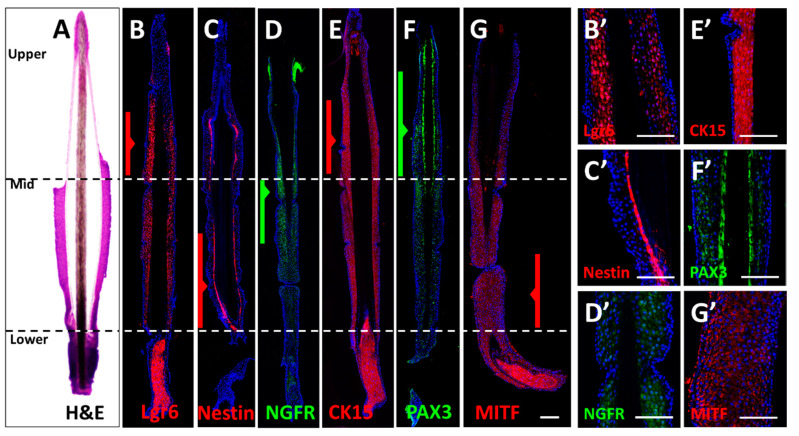
Morphological Expression of biomarkers in the plucked hair follicles. Serial longitudinal sections of plucked hair follicles were stained with H&E (**A**) and labelled immunohistochemically using antibodies for Lgr6 (**B**,**B’**, red), nestin (**C**,**C’**, red), NGFR (**D**,**D’**, green), CK15 (**E**,**E’**, red), PAX3 (**F**,**F’,** green), MITF (**G**,**G’**, red). The highest staining intensity was highlighted by broad arrows amid the distal, mid, and proximal portions of the outer root sheath (ORS), which are segmented by dashed lines. The areas with the most prominent signal have been displayed in higher magnification (B’–G’). Magnification: (**A**–**G**) sutured images, (**B’**–**G’**) 40×; scale bar 100 μm.

**Figure 2 biomolecules-11-00154-f002:**
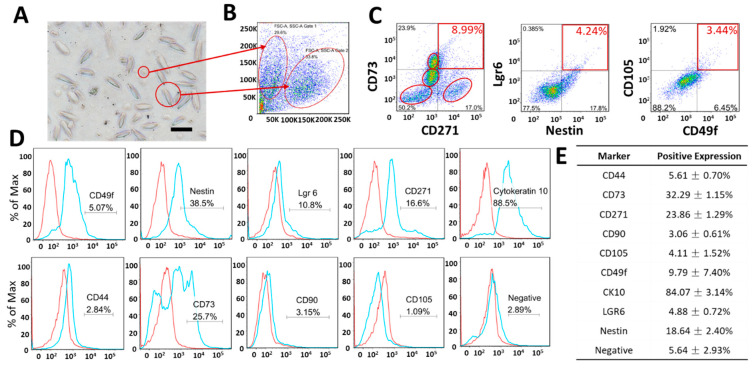
Morphological parameters and expression of CD73, CD271, Lgr6, nestin, CD105, CD49f, CD44, CD90, CK10 in a single cell suspension generated from the plucked hair follicle ORS. Control staining using isotype antibodies has been used to establish the gates for analysis. The compensation of the double fluorescence staining was <1%. (**A**) Heterogeneous cells were obtained in a single cell suspension by the means of trypsinization, and two major morphologically distinct cell populations were microscopically observed—smaller and larger cells. (**B**) Plot graph of forward and side scatter-based distribution (FSC and SSC) identified two major morphologically distinct ORS cell populations. (**C**) Fluorescence-based double-stained plots of ORS cells as representatives of stem cell populations. Double-stained cells were highlighted (red boxes). Four subgroups of cells were found in the plot of CD271/CD73 staining (red circles). (**D**) Single staining of CD49f, nestin, Lgr6, CD271, CK10, CD44, CD73, CD90, CD105, and negative MSC markers in the ORS population. Negative markers comprised a combination of the supplied negative MSC cocktail (PE CD45, PE CD34, PE CD11b, PE CD19, and PE HLA-DR). Blue histograms represent the antibody staining and the red ones represent isotype staining. (**E**) The portions of the ORS cells expressing a single marker are presented as an average percentual value ±SD. Magnification (**A**) 20×, scale bar 100 μm.

**Figure 3 biomolecules-11-00154-f003:**
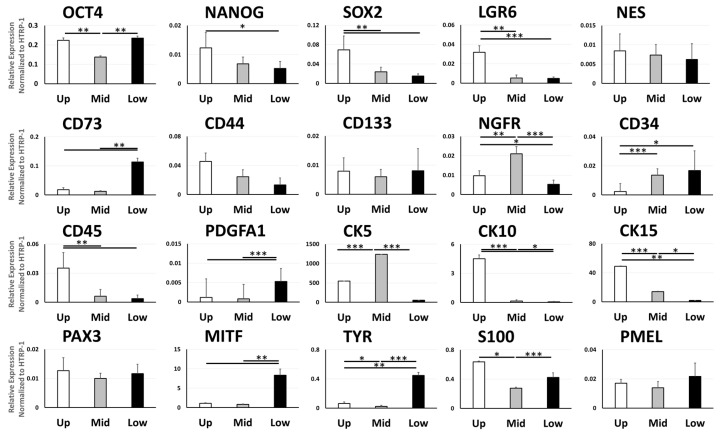
Expression of the heterogeneous gene pool in the plucked hair follicles at the level of the distal (Up), middle (Mid), and proximal (Low) ORS compartments (N = 7). Results were shown as a mean ± SEM. Comparison between groups was statistically analyzed by unpaired student’s *t*-test. Statistical significance: * *p* < 0.05, ** *p* < 0.01, *** *p* < 0.001.

**Table 1 biomolecules-11-00154-t001:** Primary antibodies used for immunofluorescent staining and flow cytometry (FCM) analysis.

Antibody	Immunoglobulin, Clone	Fluorochrome	Manufacturer
Anti-Human LGR6	rabbit IgG	Unconjugated	Sigma-Aldrich GmbH, Steinheim, DE
Anti-Human Nestin	mIgG1, clone 10C2	Unconjugated	ThermoFisher Scientific Inc., Waltham, USA
Anti-Human NGFR (CD271)	mIgG1, Clone ME20.4	Unconjugated	ThermoFisher Scientific Inc., Waltham, USA
Anti-Human CK15	mIgG1, Clone LHK15	Unconjugated	Abcam Plc, Cambridge, USA
Anti-Human PAX3	rabbit IgG	Unconjugated	ThermoFisher Scientific Inc., Waltham, USA
Anti-Human MITF	mIgG1κ, Clone D5	Unconjugated	ThermoFisher Scientific Inc., Waltham, USA
Anti-human soluble adenylyl cyclase (sAC)	mlgG1, ADCY10	Unconjugated	CEP Biotech Inc., Tamarac, USA
Anti-human Nestin	mIgG1, 10C2	Unconjugated	Cell Signaling Technology, Inc., Danvers, USA
Anti-Human CD44	mIgG2a, 156-3C11	Unconjugated	Cell Signaling Technology, Inc., Danvers, USA
Anti-Human CD34	mIgG1κ, B-6	Unconjugated	Santa Cruz Biotechnology, Inc., Dallas, USA
Anti-Human CD133	Rabbit IgG, D2V8Q	Unconjugated	Cell Signaling Technology, Inc., Danvers, USA
Anti-Human CD44	mIgG2b, Clone G44-26	PE	BD Biosciences, San Jose, USA
Anti-Human CD73	mIgG1, Clone AD2	APC	BD Biosciences, San Jose, USA
Anti-Human CD90 (Thy-1)	mIgG1, Clone 5E10	FITC	BD Biosciences, San Jose, USA
Anti-Human CD105 (ENG)	mIgG1, Clone 266	PerCP-Cy5.5	BD Biosciences, San Jose, USA
MSC Negative Mixture	CD34, CD11b, CD19 PE, CD45 PE, HLA-DR	PE	BD Biosciences, San Jose, USA

**Table 2 biomolecules-11-00154-t002:** Primer sequences for qRT-PCR analysis of marker gene expression.

Gene		Primer Sequence
*CK15*	For	AGTGGATGGACAGGTGGTTT
*CK15*	Rev	CTGATGAGAGTGGGGAGTGG
*CK5*	For	GCTGACACGAGAACCCAAAG
*CK5*	Rev	ATTGGGGTGGGGATTCTGTT
*CK10*	For	GGTGGTGGATTTGGAGGAGA
*CK10*	Rev	TCTTCCAGAGCCCGAACTTT
*TYR*	For	AGTAATGTCCAGGTTCCCAGA
*TYR*	Rev	ATGGGCTTAGGGGAAAATGTT
*PAX3*	For	CTGCGTCTCCAAGATCCTGT
*PAX3*	Rev	TTTTCTTCTCCACGTCAGGC
*MITF*	For	CAGTGGTTTGGGCTTGTTGT
*MITF*	Rev	TGACCAGGTTGCTTGTATGC
*MYC*	For	ATTCTCTGCTCTCCTCGACG
*MYC*	Rev	AGCCTGCCTCTTTTCCACA
*NES*	For	CTGCGGGCTACTGAAAAGT
*NES*	Rev	GTTTGCAGCCGGGAGTTC
*LGR6*	For	CAGGTGGAGGCTTGTCAGG
*LGR6*	Rev	TCACACTGCTGAGTTTTGGT
*NANOG*	For	CCTATGCCTGTGATTTGTGGG
*NANOG*	Rev	AGTGGGTTGTTTGCCTTTGG
*OCT4*	For	GGAGTTTGTGCCAGGGTTTT
*OCT4*	Rev	TGTGTCCCAGGCTTCTTTATT
*PMEL*	For	ACTCTTTGACTCCTCACACAGC
*PMEL*	Rev	ATTTCAAATGGGGATCATAATGT
*CKIT*	For	ATCAGCGCATAACAGCCTAAT
*CKIT*	Rev	CCAGCAAAATCAGAGTTAATCG
*CD73*	For	CTTTCGCACCCAGTTCACG
*CD73*	Rev	TCGTTGGTGTGCAAAATCGT
*CD45*	For	CTTAGGGACACGGCTGACTT
*CD45*	Rev	TGCCCTGTCACAAATACTTCTG
*NGFR*	For	GGACGCCTCGGATTCTGTAG
*NGFR*	Rev	CTTCCAGGGCATTCGGTTCA
*CD34*	For	CTACAACACCTAGTACCCTTGGA
*CD34*	Rev	GGTGAACACTGTGCTGATTACA
*HPRT1*	For	GCTTCCTCCTCCTCTGCC
*HPRT1*	Rev	CACTAATCACAACGCTGGGG
*CD44*		QT00073549, QuantiTect Primer Assays, Qiagen, DE
*CD133*		QT00075586, QuantiTect Primer Assays, Qiagen, DE

## Data Availability

The data presented in this study are available on request from the corresponding author.

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
