# Peer review of "The Middle Part of the Plucked Hair Follicle Outer Root Sheath Is Identified as an Area Rich in Lineage-Specific Stem Cell Markers"

_biomolecules, 2021, doi:10.3390/biom11020154_

Round 1

Reviewer 1 Report

This manuscript dealt with identification of hair follicle outer root sheaths as a source of stem cells via gene and protein expression. The data is systematic and comprehensive for markers. However, to identify as stem cells, differentiation is an essential criteria, not only the differential gene expression. So, if possible, the authors should provide the differentiation capacity.

Minor:

  1. For the section of 2.2. Figures, Tables and Schemes, they can be put in Results rather than Materials and Methods.
  2. Figure 1A, the biomarker labels at the bottom can be consistent with the positive immunostaining color.
  3. Figure 2A, the scale bar is missing.
  4. Figure S1Q, the scale bar is not invisible.
  5. Figure S3B, transwell migration assay was not described in Materials and Methods.
  6. Cell culture condition was not described clearly, for example, the medium, medium change frequency, seeding density, surface coating, proliferation, and so on.

Author Response

Dear Editor,

Dear Reviewers,

Thank you very much for your fast review of our manuscript and your valuable comments on our manuscript.

We tried to fulfill all the reviewers’ requirements herewith improving the manuscript and rearranging the figures accordingly. We kept the traking mode, and highlighted the major changes in the manuscript in yellow, and we hope these help your revision.

Please find all further concerns addressed below by our point-by-point answers to the reviewer’s comments.

Reviewer 2 Report

In the study, Hanluo Li and colleagues, revealed that the middle part of the hair follicle outer root sheath (ORC) holds potential of yielding multiple stem 40 cells, in particular mesenchymal stem cells. The manuscript is well written and interesting, but there are some unconvincing points.

Major concern

In Figure 1, the signals of Lgr5 and MITF are indistinguishable in the middle and proximal regions. Also, the signal levels of Lgr6 and CK15 seem to be the same in all three regions.

How can we recognize these differences?

I think FACS data should be populated to recognize the differences in the properties of stem cells in the proximal, middle, and distal regions.

The title presents the presence of multiple stem cells in the middle part of the ORC. I believe that the multipotency of these cells needs to be proven by in vivo or in vitro experiments.

Author Response

(The authors gave the same response as above.)

Reviewer 3 Report

General Comments

The authors examined the outer root sheath of human scalp hair follicles for potential stem cells.  The study has clearly indicated that multiple cellular populations reside in the ORS.

Specific Comments

Methods

Hair donors; the authors need to provide more information about donors such as gender, age, and examination for potential problems associated with normal hair growth.  Gender assignment is important as male hair growth is regulated by hormones differently that female hair growth.  Age also plays a role in regulating hair growth.  Also, were equal numbers of hair follicles obtained from each donor and how these hairs distributed among the various assays.

Discussion

This section is overly long. The material discussed is relevant; however, it could be more succinctly shortened.

General Comments

The authors examined the outer root sheath of human scalp hair follicles for potential stem cells.  The study has clearly indicated that multiple cellular populations reside in the ORS.

Specific Comments

Methods

Hair donors; the authors need to provide more information about donors such as gender, age, and examination for potential problems associated with normal hair growth.  Gender assignment is important as male hair growth is regulated by hormones differently that female hair growth.  Age also plays a role in regulating hair growth.  Also, were equal numbers of hair follicles obtained from each donor and how these hairs distributed among the various assays.

Discussion

This section is overly long. The material discussed is relevant; however, it could be more succinctly shortened.

Author Response

(The authors gave the same response as above.)

Round 2

Reviewer 2 Report

Since you answered the first revise, I think you should accept.